# Exploring proxies for occupation intensity in hunter-gatherer settlement systems: A combination of ethnohistoric and archaeological data

Amy E. Clark[1]*, Guadalupe Sánchez Miranda[2], Neftalí López-Pérez[3], Antonio López-Rivera[3], Tamara Luna[3], Astrid Avilés[2], Richard Martynec[4], Sandra Martynec[4], Natalia Martínez-Tagüeña[5], Matthew Pailes[6]*

1 Department of Anthropology, Harvard University, Cambridge, Massachusetts, United States of America, 2 Instituto Nacional de Antropología e Historia, Hermosillo, Sonora, Mexico, 3 Escuela Nacional de Antropología e Historia, Ciudad de México, Mexico, 4 Independent Researchers, Tucson, Arizona, United States of America, 5 Centro de Investigación en Ciencias Sociales y Estudios Regionales, Universidad Autónoma del Estado de Morelos, Cuernavaca, Mexico, 6 Department of Anthropology, University of Oklahoma, Norman, Oklahoma, United States of America

* amy_clark@fas.harvard.edu (AEC); mpailes@ou.edu (MP)

## Abstract

A primary concern for hunter-gatherer archaeology is whether occupation intensity can be broken down into its constituent components: group size, length of stay(s), and frequency of reoccupation. This article contributes to this discussion with settlement pattern data from the traditional homeland of the Hia-Ced O'odham. We employ multiple material proxies of occupation intensity in addition to site area. Our approach highlights that patterns produced by logistically mobile systems with significant levels of site reuse present unique obstacles that contrast with the residentially mobile systems that underpin much current discussion and most ethnographic baselines. We provide one simple measure for identifying the relative magnitude of site reuse in settlement pattern data. Our multiple proxy landscape scale analysis also allows us to move beyond broad characterizations of economic strategies and identify site specific roles and strategies within larger settlement systems. Rather than viewing sites with anomalous relationships between proxies as problematic, they provide an avenue for identifying unique components of settlement systems and the impact of social negotiations intrinsic to human landscape use.

## Introduction

A central issue in hunter-gatherer studies is documenting the range of variation in mobility strategies and how these were constrained by ecological, demographic, and historical factors [1]. Our understanding of the relationship between mobility parameters (e.g., length of stay, group size, and site reuse frequency) and these

**Data availability statement:** Data from the Sierra Pinacate are included as a supplementary document as are utilized data from Las Playas and Coastal sites. Site locations are considered sensitive and not released with precision to avoid vandalization.

**Funding:** GS; 2021-007; Agnese Nelms Haury Charitable Trust of the University of Arizona; https://www.haury.arizona.edu/ GS; NA; Centro-INAH-Sonora; NA.

**Competing interests:** The authors have declared that no competing interests exist.

constraints primarily come from ethnographic sources [2–7]. These studies predict how hunter-gatherer groups structure their mobility in a given environment but do not provide reliable material correlates to reconstruct these patterns in archaeological contexts. Even for studies with an explicitly materialist (i.e., archaeological) focus, where attention is paid to the objects and things left behind, the sites under study are not archaeological. They had not been subjected to various formation processes including the reuse and repurposing of a site and its objects by other humans across many subsequent visits. As a result, ethnographic observations often capture only one or two of the contributing components of occupation intensity. Archaeological applications often expressly focus on discriminating site histories on a continuum of site reuse contrasted to extended occupation. Rarely can these site-centric studies situate settlements within a broader pattern of landscape use. Even when multiple sites are considered, they often cannot be assumed to be exhaustive of peri-contemporaneous site types and thus reflective of a larger mobility strategy [e.g., 8,9]. This is particularly problematic when archaeological proxies of occupation intensity at one or a few sites are interpreted as landscape scale patterns of mobility, most often parsed in Binford's forager (residential mobility) and collector (logistical mobility) land use spectrum [3,10]. In this study, we explore the use of multiple proxies of occupation intensity and their deviations from each other to provide a template for interpreting the roles of different sites in a landscape scale system and how these combine to produce typical hunter-gatherer settlement systems.

We consider three proxies for occupation intensity: site area, the number of wind-breaks (structures), and the number of ground stone artifacts. Although ground stone is not ubiquitous in hunter-gatherer archaeological sites, its utility as a proxy should be similar to other artifact classes that have long (curated) use lives. Likewise, wind-breaks function as a proxy similar to other features, such as hearths. While these particular proxies may not be universal, they provide a template for how the exploration of multiple classes of artifacts/features adds nuance to landscape scale investigations of occupation intensity. Important for our study, these are proxies quantifiable from survey data conducted at landscape scales.

## Theoretical background

Investigations of occupation intensity leverage proxies such as site size, counts of material, and/or number of features to estimate the relative amount of cumulative person hours expended in different locations and settings. Person hours are understood as resulting from group size and group length stay, with the latter further divisible into length of individual stays and number of stays [8,11–13]. Much progress has been made in the development of methods to parse variation in the average length of site occupation, which is often interpreted through the lens of residential vs logistical mobility. In a residential mobility strategy, most resources are acquired within a day's journey of the residential camp, so that when these resources are exhausted, the residential camp is moved to a new location. Different ecosystems variably encourage the reoccupation of the same base camps repeatedly or more random placement. In a logistical mobility strategy, resources are often acquired within a much larger

radius so that logistical camps are established for short term stays to supply the base camp. This results in longer duration base camp occupations that tend to be reliably reoccupied seasonally. There is no clear cut division between these two strategies and most prior archaeological applications focus on some aspect of lithic assemblages to discern placement on a continuum between these two strategies [13–18]. Additionally, despite their clear conceptual differences short term logistical camps and short term residential base camps may look quite similar archaeologically. Many of the above cited studies evaluate patterns at a single or a limited number of extensively excavated sites, making it difficult to assess diversity among sites at a landscape scale. This has the effect of collapsing questions of landscape scale mobility patterns and site level studies of occupation intensity into a single dimension of long stays = logistical and short stays = residential. This approach is both limited in practical application as it requires substantial assemblages from presumptive base camps and limits interpretative nuance in its fixation on one dimension of occupation intensity.

At the landscape or settlement system scale, ethnological and more rarely archaeological studies have explored the fidelity of potential proxies, particularly site area, in measuring components of occupation intensity [7,11,19–22]. These studies suggest that hunter-gatherers in a residential mobility system occupy camps for periods of time that conform to power-law distributions [23], providing a baseline expectation of the distributional character of occupation intensity proxies. Recent applications of Settlement Scaling Theory add nuance to this general pattern by demonstrating hunter-gatherers produce less dense sites as populations grow. Stated alternatively, site areas grow more quickly than populations as both increase in hunter-gatherer settlements. This contrasts markedly with modern and ancient sedentary populations that require per capita less space as populations increase [19,24–26]. This indicates that fundamentally different dynamics characterize the value of social interaction within hunter-gatherer settlements relative to sedentary societies. The value of additional social interaction appears to rapidly depreciate beyond that of a typical band sized unit and often that of a nuclear family owing to a lack of social mechanisms to ensure returns on individual investments [20,26,27]. In other words, when hunter-gatherers aggregate in larger numbers, they employ larger spatial buffers to retain residential unit structure and limit sharing networks. Though these lines of study are highly productive, they are not readily applicable to the vast majority of the hunter-gatherer archaeological record in which site reuse is a fundamental component of occupation intensity. Ethnographic studies often consider a selectively chosen subset of settlements very different from the samples recorded in archaeological surveys. The studies cited above focus on highly residential mobile groups or the portion of the annual cycle that corresponds to high residential mobility and typically measure only the momentary use of space—not the cumulative use of space visible in material remains. Archaeological applications of Settlement Scaling Theory similarly invest significant effort to ensure proxies only capture the momentary maximum of population size. This is a very different concept than occupation intensity that explores the structure of cumulative time spent at locations.

It is far more common for archaeologists to undertake research intended to establish residential vs logistical mobility patterns or otherwise confront data sets that cannot be preemptively winnowed to capture only the residential mobility base camp component of settlement systems. The modern and historic ethnographic sample of hunters-gatherers also may be skewed toward high residential mobility groups not characteristic of most past hunters-gatherers [28]. In short, even when ethnographic and archaeological investigations use the same proxy (settlement size) they may be measuring different phenomena when the settlement system in question does not pertain to the extreme residential end of the logistical/residential mobility spectrum. Our discussion thus impinges upon, but is not intended to critique cross-cultural studies that explore the relationship between site area and demography such as those proposed by Settlement Scaling Theory [25]. It is certainly possible to explore these and other generalizable relationships through careful data hygiene that eliminates cases where settlement reuse is less of an issue and where a clear demarcation between base camp and other types of sites is evident. This, however, excludes most of the hunter-gatherer archaeological record and limits interpretations of larger landscape scale use.

Our study advances along an alternative line of investigation largely instigated by Haas et al. [29]. This study demonstrated that archaeological proxies of occupation intensity such as artifact inventories (projectile points) also present

power-law distributions but critically note that, in contrast to ethnographic studies, site size as measured by artifact dispersions do not scale in the same way or predictably, presumably due to groups following variable logics of space reuse at different site types in the larger settlement system. This power-law signature in material proxies plausibly arises from a preferential attachment rule (the rich get richer) [30] in which groups revisit past utilized locations due to the concentration of materials or established symbolic importance [31]. It is not surprising that power-law distributions can be identified in both archaeological and ethnographic contexts despite differences in the underlying processes. This is because the product of multiple power-laws distributions is again a power-law distribution [32:235] (i.e., individual occupation episode length * propensity for reuse). Failure to find power-law distributions in archaeological hunter-gatherer site size distributions that cannot be pre-emptively selected to reflect momentary maximums of occupation suggests site reuse is a fundamental component of occupation intensity.

This study builds on previous efforts to overcome some of the above noted challenges. Specifically, we employ established methods to affirm the lack of power-law distributions in site areas and their presence in artifact inventories. We then employ linear regression to explore the underlying causes of these results by exploring how material correlates depart from site areas and each other. In contrast to many past investigations of occupation intensity, we employ material correlates recorded at a landscape scale to understand the entirety (or at least a greater part) of an integrated hunter-gatherer economy. In our approach, the lack of consistent correlation between archaeological site sizes and artifact inventories is seen as an opportunity to provide finer grained interpretations of site function and landscape use strategies reflective of social and economic constraints and opportunities. Our approach tacks between broad theoretical precepts and context specific data provided by a minimal but valuable ethnohistoric record. This exercise serves to both test the applicability of broad theory and to contribute to its refinement by invoking historically contingent variables when expectations are not met. Like most anthropological investigations, the approach alternates between deductive and inductive modes of explanation. We do not intend this article to be read as a cautionary tale of how historical contingency renders abstraction and generalization difficult but rather seek to both enrich our understanding of a particular context while offering insights on factors relevant to other datasets with anomalous cases.

## Environmental and cultural background

The data utilized for this study derive from a recent archaeological survey of sites in the Sierra Pinacate region of Sonora, Mexico, and legacy data from adjacent ecological settings all pertaining to an area ancestrally occupied by the Hia-Ced O'odham, a hunter-gatherer-fishing group. This larger region is known as the Gran Desierto, itself a sub-region of the Sonoran Desert, and includes coastal settings along the upper Gulf of California (Fig 1). Temperatures regularly exceed 40 C with annual precipitation totals less than 90 mm [33]. These conditions make the availability of water a fundamental constraint on mobility patterns. Within the Gran Desierto, the Sierra Pinacate, comprised of geologically recent volcanic features, includes numerous canyon channels that hold water in bedrock tinajas following rains. A few locations hold water for most of the year and all tend to be predictable in volume relative to the last rainfall event [34]. Elsewhere in the region water can be obtained through hand dug wells in near surface contexts at localized fresh water springs in the salt flats of Bahía Adair and at locations along the dry course of the Sonoyta River [35,36].

The materials described here pertain predominantly to the proto-Hispanic and antecedent late Ceramic period (post CE 800) with some sites evidencing use since the Archaic (~5000 BCE). The overall demographic trajectory of the region implies a sparse population prior to the Ceramic Period [37]. Padre Eusebio Kino and his affiliates Juan Mateo Manje and Juan María de Salvatierra provide the first appreciable European observations of the region on four explorations from 1698 to 1706 [38]. Specific locations mentioned on these journeys correspond with sites revisited in our archaeological survey. Oral histories collected in the mid 20[th] century note a distinction between Hia-Ced O'odham bands that pursued fully hunter-gatherer-fisher economies and those that engaged in irregular *temporal* agriculture [39,40]. Tensions existed between these groups with the fully non-horticultural "Pinacateños" maintaining closer relationships with Yuman/

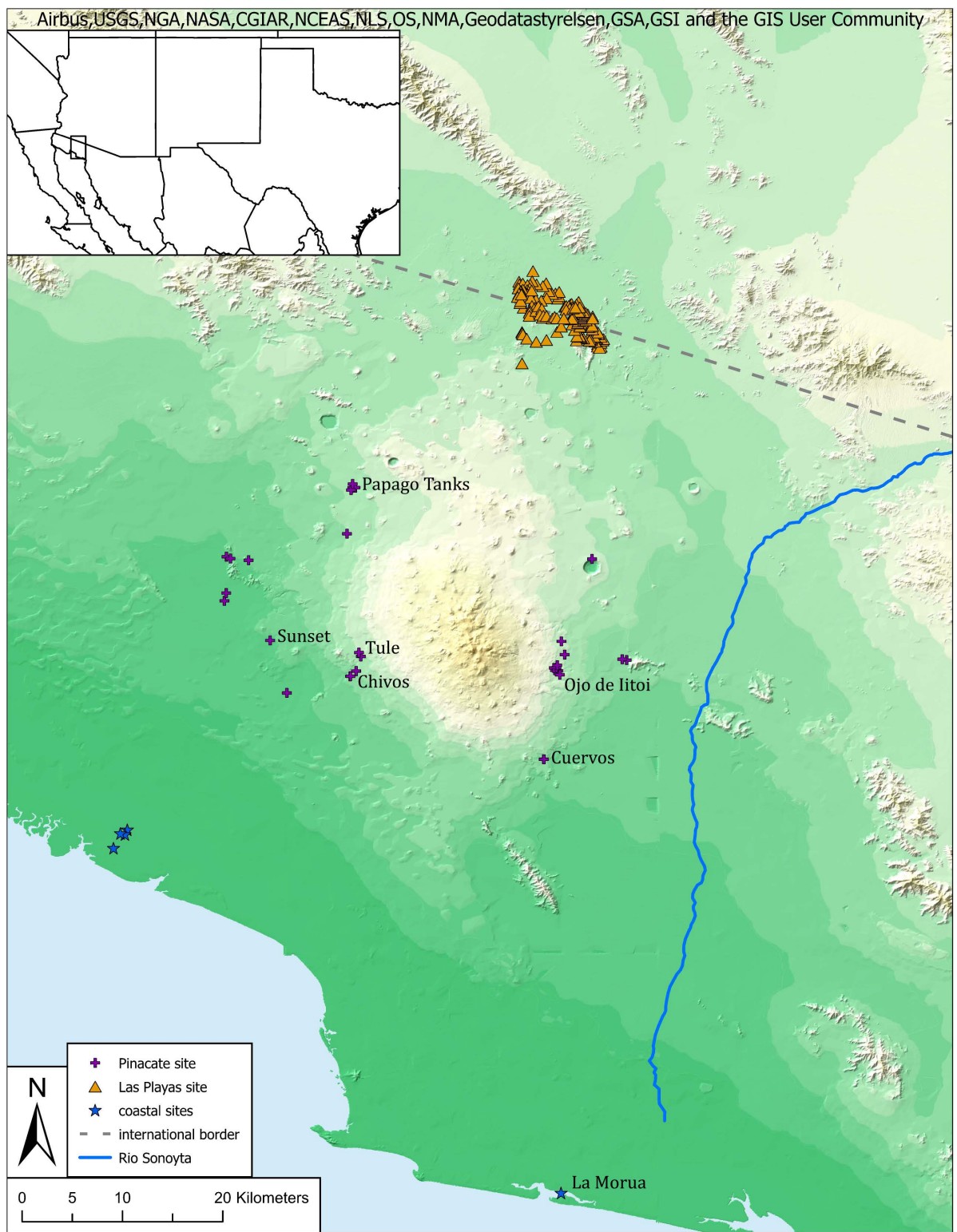

Papago Tanks

Sunset

Tule

Chivos

Ojo de Iitoi

Cuervos

La Morua

N

| | Pinacate site |
| --- | --- |
| | Las Playas site |
| | coastal sites |
| | international border |
| | Rio Sonoyta |

0    5    10         20 Kilometers

**Fig 1. Study region.** The Sierra Pinacate situated in the Gran Desierto. Portions of image are the intellectual property of Esri and is used herein under license. 

Patayan groups of the Colorado River delta to the west [41] than their linguistic O'odham relatives. Hia-Ced presence in the broader region is only sporadically documented by Europeans throughout the later 1800s but there is an unequivocal record of persistence as a component of modern O'odham identity [42,43].

The historically recorded Hia-Ced residents in the Gran Desierto were paragons of a subsistence economy that vacillated between forager and collector modes. Their range included diverse ecological zones of coast, estuary, dunes, bajadas, playas, and river course. Extended seasonal stays at reliable tinajas of the Sierra Pinacate anchor the logistical end of this spectrum while other contexts supported periods of high residential mobility. Ethnohistoric estimates of the Hia-Ced Pinacateño population range from 100 to 500 individuals, with a typical group comprising 20 individuals [44:62]; similar to group sizes commonly observed among residential mobility hunter-gatherers [20,45]. Larger gatherings were facilitated in times of resource and water abundance that were marked by ceremonial activities. Water was the most constraining resource and dictated the timing of location use, the relative degree of logistical organization, and the degree of inter-group diversity in land use strategies feasible in any given season [40,41,46,47]. The temporally and spatially limited availability of tinajas led to repeated reuse of adjacent camps; this produced a record ideal for evaluating site reuse contributions to occupation intensity at the landscape scale.

Neighboring agriculturalist/horticulturalist Akimel and Tohono O'odham oral histories recount their regular traversal of the region on salt pilgrimages to the Bahia Adair region. Hayden [41:341] noted terse but apparently permissive inter-O'odam encounters in the Sierra Pinacate between resident hunter-gatherer groups and those on ritual salt journeys and more mundane resource acquisition ventures. There is thus ethnohistoric documentation of multiple bands that made regular use of the Sierra Pinacate as well as other O'odham groups who periodically entered the region and evidence for contacts with exterior groups such as the Patayan/Yumans to the west. We emphasize this apparent tolerance of exterior groups in the region and common pool resource status of some locations [48] due to relevance in subsequent interpretations.

## Archaeological background

The avocational archaeologist Julian Hayden undertook an extensive survey of the Sierra Pinacate between 1958 and 1987. Hayden's interests centered on earlier time periods, particularly the purportedly pre-Clovis Malpais culture. Hayden's [49] explorations of the Sierra Pinacate also noted later archaeological materials, identifying 23 tinajas with cultural materials as well as 19 other locations. Other archaeological research in the region is limited but highly valuable to our discussion. To the immediate north of the Sierra Pincate, a survey of 64 km$^2$ located 159 sites [50,51]. This "Las Playas" region includes three basins that seasonally fill with high inter-annual variability. There are important differences between the sites of the Las Playas area and the focus of our recent efforts in the Sierra Pinacate. These include the apparent minimal use of wind breaks and the manufacture of shell jewelry for export, inferred by the prevalence of *Laevicardium* (sp) co-occurring with shell-working tools, the latter absent in the Sierra Pinacate. A significant majority of Las Playas sites are small, presumably short term camps, some of which nonetheless evidence sporadic use across centuries to millennia.

Along the coast south and west of the Sierra Pinacate, investigations of shell midden sites capture another facet of subsistence economies [52,53]. These shell-midden sites evidence periodic use extending to the Middle Archaic [54]. More recent deposits include a diversity of ceramics from the Patayan, Hohokam, and Trincheras regions, possibly reflecting multi-cultural use of the area but more plausibly the extra-regional connections of local ancestral Hia-Ced O'odham. The prevalence of mollusk species in shell middens reflects subsistence activities that have minimal overlap with the shell jewelry inventories of neighboring agricultural groups. These middens span an incredible size range from discrete concentrations a few m$^2$ in area to the largest, La Morúa, at ~4.6 million m$^2$.

## Data collection and methods

Our collaborative binational effort was intended to begin the task of cataloging Hayden's site locations for registration in INAH (*Instituto Nacional de Antropología e Historia*) site files, a need made more pressing by the region's designation

as a UNESCO World Heritage Site. Our research was conducted in accordance with Mexican regulations and approved by the Consejo de Arqueología and in coordination with the *Reserva de la Biosfera El Pinacate y Gran Desierto de Altar.* All necessary permits were obtained for the described study, which complied with all relevant regulations. Data collection focused on relocating sites and thoroughly recording a sub-set of these. We also recorded a minimum number of newly identified sites along routes taken to or from known locations.

Following INAH standards, survey teams designated sites based on the presence of features and/or the co-occurrence of at least three artifact/ecofact classes: ceramic, chipped stone, ground stone, marine shell, and faunal remains. Once a site was identified, its boundaries were demarcated by the approximate limits of easily identified surface artifacts and features. There were significant differences in material densities within and between sites. Crews endeavored to demarcate sites through a systematic approach in which sites boundaries capture the threshold at which materials are no longer easily intervisible at ground level. This is the defacto method of site boundary definition in most Sonoran Desert contexts in both US and Mexican institutions, rendering our results comparable with previous surveys. Nearly all ground stone and features were photographed and point plotted with a Garmin GPS. Larger sites were subjected to drone facilitated aerial survey. In several cases, this led us to expand the boundaries of sites to include features missed on ground level survey. Ground stone attributes were determined from the photographic record aided by 51 in-field measurements of implement depth. Large sites were sub-divided into loci. We conduct most statistical analysis using loci but present site level analyses in the supporting documents to monitor whether definitional criteria overly-determine interpretations (S1 Supporting Information, S1-S4 Figs in S1 File) [55].

Our Pinacate settlement pattern data incorporates sites spanning the site size hierarchy with a particular emphasis on diversity among larger sites. Isolated shell concentrations, lithic scatters, and rock features abound throughout the Sierra Pinacate and our results do not provide a representative frequency of smaller sites or isolated features and artifacts. To improve the representativeness of the sample for certain analyses we incorporate the data from the Las Playas region [50,51]. Identical to our Sierra Pinacate survey, the Las Playas effort systematically noted features, including wind breaks, which were very rare, as well as more abundant ground stone implements. These observations facilitate amalgamating the two samples, but the details of ground stone observation are cruder in the Las Playas data. A third component of the subsistence economy was anchored to littoral resources [52,56]. Recent work in this region has focused on larger or unique sites [53] and is not amenable to a separate distributional analyses. Systematic artifact counts are lacking except for a few sites and described with a less specific typology. We cannot incorporate these sites into most of our analyses but consider them where feasible and in qualitative interpretations.

Temporally sensitive diagnostic artifacts (projectile points, ceramics, etc.) across all samples reflect significant time averaged deposits, resulting from variable degrees of site reuse. Many sites were likely utilized only a few times, potentially across hundreds of years. Others reflect much more intensive use spanning from Archaic (but mostly late Ceramic) to Historic times. The only sites omitted from consideration are those lacking relevant data or that clearly correspond to historic contexts not-indicative of a hunter-gatherer-fisher economy.

## Proxy quantification

Ground stone implements (Fig 2) were used to grind seeds and edible pods from leguminous trees (e.g., *Prosopis, Cericidium,* etc.) and thus closely track subsistence consumption and attendant cumulative demographic presence, i.e., occupation intensity. We explored several options to quantify ground stone that took into account fragmentation and relative degrees of wear (see S1 Table in S1 File), but found an extremely high degree of correlation between these approaches and a simple count of "bottom stone" (nether stone) [57] implements and thus employ this more intuitive approach to quantification. Bottom stone counts should be more reliable than entire inventory counts as perishable wooden pestles were commonly employed in place of top stones (hand stones). Only for the Coastal site inclusive sample do we instead consider all major ground stone implements as typological specifics are not reported. We utilized a simple typology for

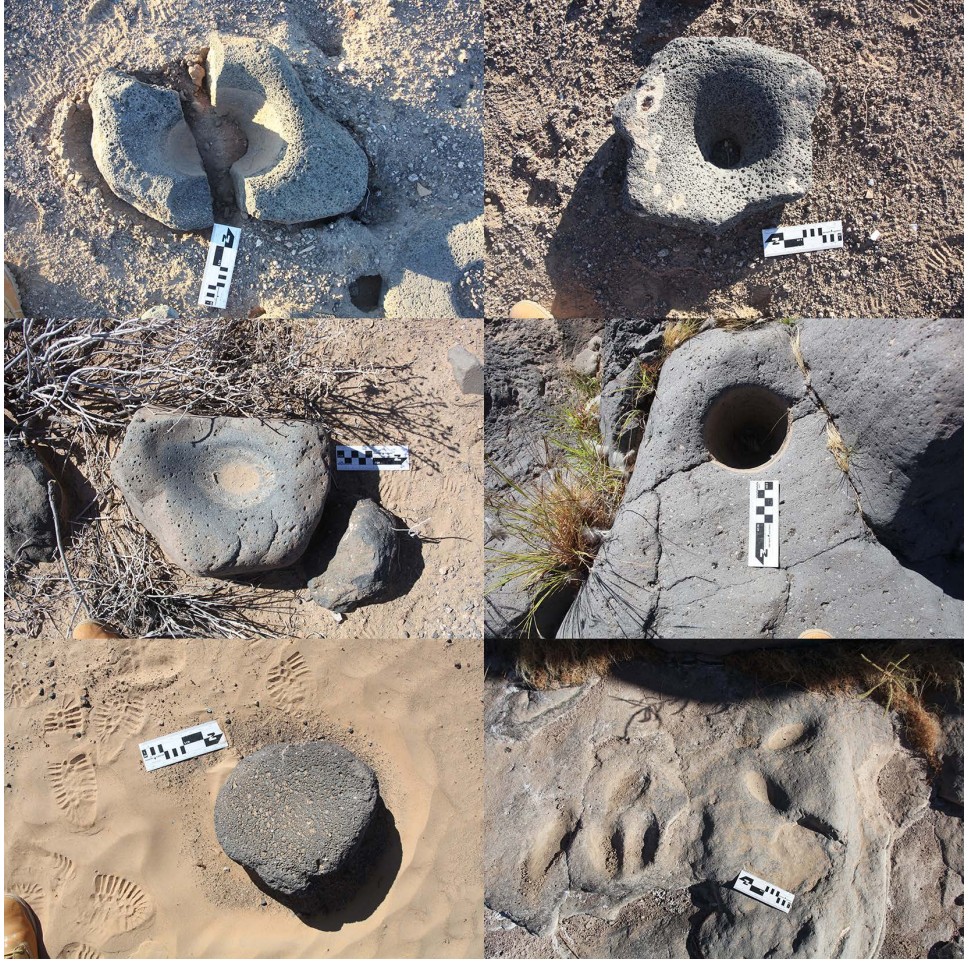

**Fig 2. Ground stone.** Clockwise from top left, three mortar fragments, complete gyratory crusher, bedrock mortar, grinding slicks, flat bottom stone, basin metate.

rock features: structures, concentrations, and figures. Structures or "sleeping circles" correspond to arcs or straight lines that uncontroversially indicate windbreak foundations (Fig 3). Concentrations are amorphous clusters of rock that appear anthropogenic but lack any clear interpretation. The vast majority reflect windbreaks that were partly disassembled to reuse constituent stones in new constructions or windbreaks disturbed by other anthropogenic processes. Rock figures (Fig 4) correspond to geoglyphs, demarcated spaces, and other alignments insufficiently robust to have served as wind-breaks. We combine counts of structures and concentrations but not figures in the calculation of "total structures" in our analysis of Sierra Pinacate sites to serve as a second material proxy of occupation intensity.

## Description of surveyed sites

To develop explanations regarding deviations in the expected relationship between proxies, it is necessary to provide considerable contextual details on a few sites and several groups of sites (Fig 1). The sites discussed individually below correspond to locations that we interpret as logistical base camp or special use camps.

Papago Tanks is located at the northern end of the Sierra Pinacate region. This site encompasses the most reliable and often year-round water source. In our investigations, undertaken in a particularly dry year near the end of the dry season,

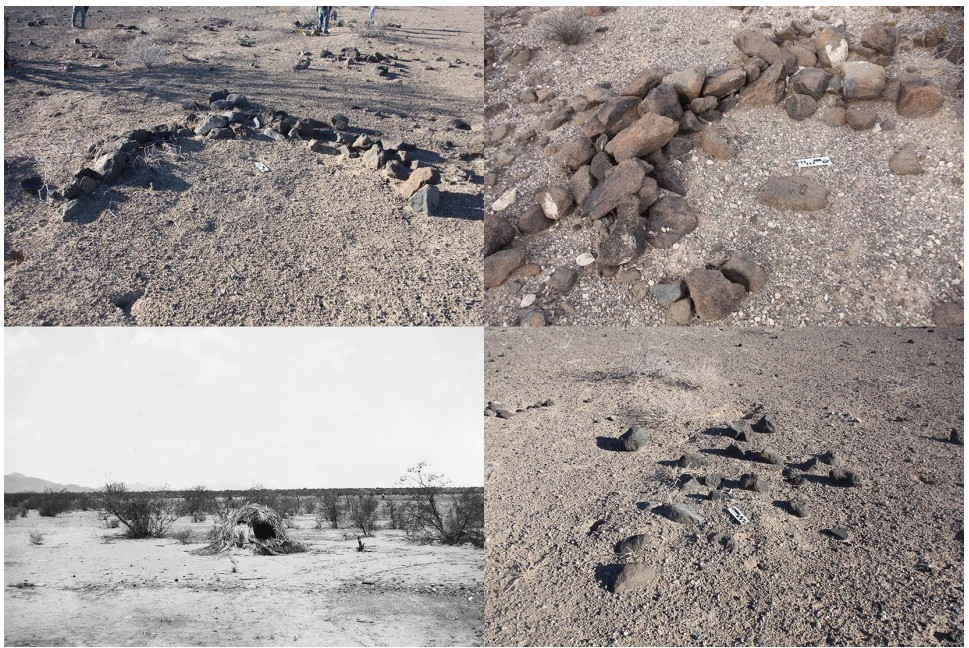

**Fig 3. Windbreak structures.** Top row, examples of preserved wind break structure foundations, bottom right example of rock concentration indicative of disturbed-partially deconstructed wind break, bottom left ethnographic photo of O'odham ephemeral structure: Museum of Cultural History, University of Oslo, Norway, UEMf03034-CL-NBR9213.770.

there was still appreciable water only here and at the tinaja adjacent to the Chivos site. This location was selected as a hermitage by Juan Caravajales, the last individual recorded to pursue a traditional Hia-Ced subsistence economy [58]. Within the tinaja channel there were numerous grinding slicks and deep bedrock mortars (~20 cm). The latter is indicative of many person hours of processing labor given the wooden pestles utilized by the Hia-Ced. We also identified one of the largest rock figures on our project at this site, an oval form with dimensions of 22 by 43 m (Fig 4). Most windbreaks were located some distance from the tinaja along an adjacent north-south drainage. The site is divided into three loci: the ancillary drainage, the tinaja area, and an area to the south of the main drainage with dispersed artifacts and features including the oval feature. In contrast to other sites, Papago Tanks suggests some degree of complementarity as opposed to redundancy in the features common to each loci.

Ojo de Iitoi contains the greatest number of artifacts and features. The site is divided into five loci largely redundant in features and artifact types but highly variable in size. There is some temporal partitioning between the various loci, with most identified pre-Colonial Hohokam (CE 800–1100) ceramics present in Locus 1. We suspect loci 3, 4, and 5 to be nearly coeval in age except that locus 3, the largest, shows the greatest evidence for continual reuse and alteration with a diversity of ceramics from multiple time periods and several Archaic projectile points.

The Chivos site is split between two loci with a drainage and approximately 450 m distance between them. This site was one of several locations visited by Padre Kino and company [44]. Both loci present clear evidence of repeated reuse of stones in windbreak constructions. The two loci are linked by clear trails, albeit with modern damage from cattle. There are isolated rock figures between the loci as well as a very low density of shell and lithic artifacts.

Tule is located further up the slopes of the Sierra Pinacate from Chivos. The site was somewhat perplexing in its haphazard distribution of features and artifacts. The site is mostly located on the northwestern side of an arroyo with a smaller locus 2 composed of an artifact scatter on the southeastern side. A very large *avenida* (wide linear cleared area in desert

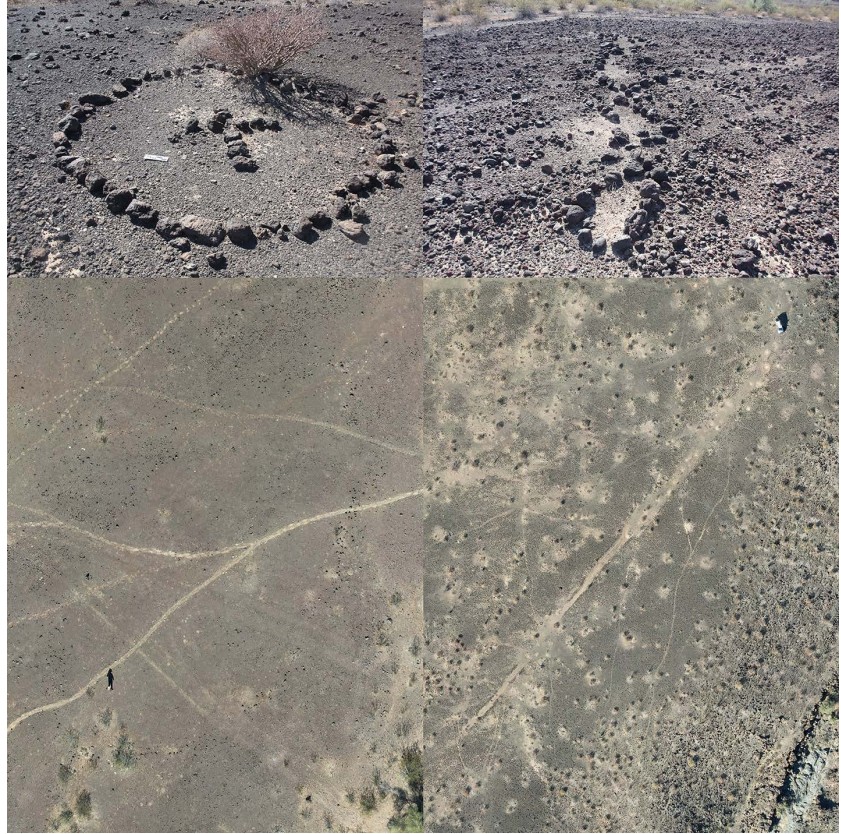

**Fig 4. Ritual rock figures.** Clockwise from top left, rock figure from Cuervos, rock figure from Tule, Grand Avenida from Tule (note truck in upper right), large outlined oval from Papago Tanks (note human bottom left).

pavement) is almost certainly indicative of specialized ceremonial activities as are several additional rock figures. The area has been affected by cattle trails and wild burrow wallows, but the numerous trails, clearings, and rock figures at the site reinforce the interpretation that unique activities were conducted at this location.

Cuervos is adjacent to the only significant tinaja on the southern margin of the Sierra Pinacate. The site is a continuous distribution of artifacts but with some internal segmentation. Rock figures including Christian crosses (Historic in age) (Fig 4), a few structures, and appreciable artifacts are present in a wide level area of desert pavement. A dense area of ground stone and other artifacts is present on the eastern margin of the site on a slightly lower topographical surface. Beyond this is an area of structures near the edge of the arroyo canyon. This site is referenced in ethnohistoric texts as pertaining to both Hia-Ced and other O'odham group use [41:341].

Sunset is one of the few substantial non-tinaja associated sites. The site is adjacent to a small playa surrounded on three sides by a dune formed over a basalt outcrop. Several structures are preserved in the playa but most material and features are located on the dunes, where constant reworking by deflation and sedimentation makes feature identification difficult. Proto-colonial artifacts, such as glass arrow points, confirm this location's association with late period occupation and there is general consensus that this was a ritual center of the Hia-Ced [58:396] well into the 1800s. There are portions of trails preserved between this site and the relatively nearby Chivos site as well as a high hill with a large cairn, potentially described by Kino as the point from which he ascertained Baja California was a peninsula [38:283].

The remaining sites are fairly non-descript lacking in clear significant concentrations of artifacts or features. Rock structures, concentrations, and rock figures are present at many of these sites but are dispersed as are ground stone implements and other artifacts. The site of Galletal is particularly illustrative in that it presented no clear structures or figures, but at least three bedrock mortars that were near 40 cm in depth, reflecting an immense amount of person hours. Many of these sites were likely visited repeatedly, especially those near less reliable/smaller but still important tinajas. These sites are clearly distinct from the above enumerated sites that served as either logistical base camps, special purpose sites, or both. Many of these qualitatively appear to reflect more haphazard, dispersed, and minimal structured use of space. Lastly, there are a number of small sites or loci that reflect short term camps or resource extraction locations that are very similar to the vast majority of sites in the Las Playas region.

## Analysis and discussion

For clarity, we begin with a definition of the relevant data sets in our analysis. The Pinacate dataset consists of sites recorded in the Sierra Pinacate (n = 27 [as loci] or n = 17 [as sites]). The Las Playas data includes 144 additional sites and the Coastal data inclusive sample an additional 6. We generally report only loci based data and amalgamations of Pinacate plus Las Playas but include alternative iterations in the supporting information for further transparency (S1-S4 Figs in S1 File). Our analysis work flow consists of three stages. We begin by examining the distributions of our ground stone proxy data. This step is solely intended to demonstrate that our data meets the expectations of a representative sample of a hunter-gatherer settlement pattern, conforming to a power law distribution. We next articulate the relationship between artifact counts (ground stone) and feature counts (structures), to demonstrate their relative strengths as alternative proxies of occupation intensity and identify points of departure relevant to producing more nuanced interpretations. Finally, through various approaches to linear regression we advance our understanding of the relationships between artifact/feature count data and the size of sites as estimators of various contributions to occupation intensity at a landscape scale.

### Assessing the representativeness of the settlement pattern

To evaluate the relationship of ground stone to previous discussions of occupation intensity, we want to ensure that our sample corresponds to an expected frequency distribution of sites characteristic of hunter-gatherer systems. This step is analogous to statistical analysis that first ensures a normal distribution is present as a sign of representative sampling, except in our case proxies of occupation intensity should correspond to non-normal distributions, with power-laws forwarded as the predictable outcome of scenarios in which cumulative previous site use positively weights the probability of future site use. This logic is especially salient for ground stone, as its established presence would encourage reuse of the same location. We do not provide an analogous exploration of feature data (structures) as this would only be available for the limited Pinacate sample. We employ the statistical methods outlined by Haas et al. [29] to test the distribution of ground stone against alternative discreet interval distributions. The underlying logic of this approach is provided by Clauset [59] with R script documentation by Gillespie [60] and its derivation in Haas et al. [29: supporting information].

The evaluation proceeds in several steps. Fig 5 presents cumulative mass functions of the ground stone counts from the various sample data sets with logarithmic axes (see also S3 Table in S1 File). This is the traditional approach to evaluating power-law structure, as such distributions produce linear plots. The data appear passingly power-law in character, but merit further investigation as visual inspection alone can be unreliable in differentiating between heavy tailed distributions [59].

Applying the Maximum Likelihood Estimation (MLE) test employed by Hass et al. [29], we determine the best fit parameters to various possible distributions appropriate for count data. These include power-law, Poisson, and geometric. These models are then tested through a goodness of fit approach. For each dataset $i$ of $n_i$ sites, the KS (Kolmogorov-Smirnov) distance is determined between the empirical data and MLE model, giving $D_m$. Subsequently, a random sample of $n_i$

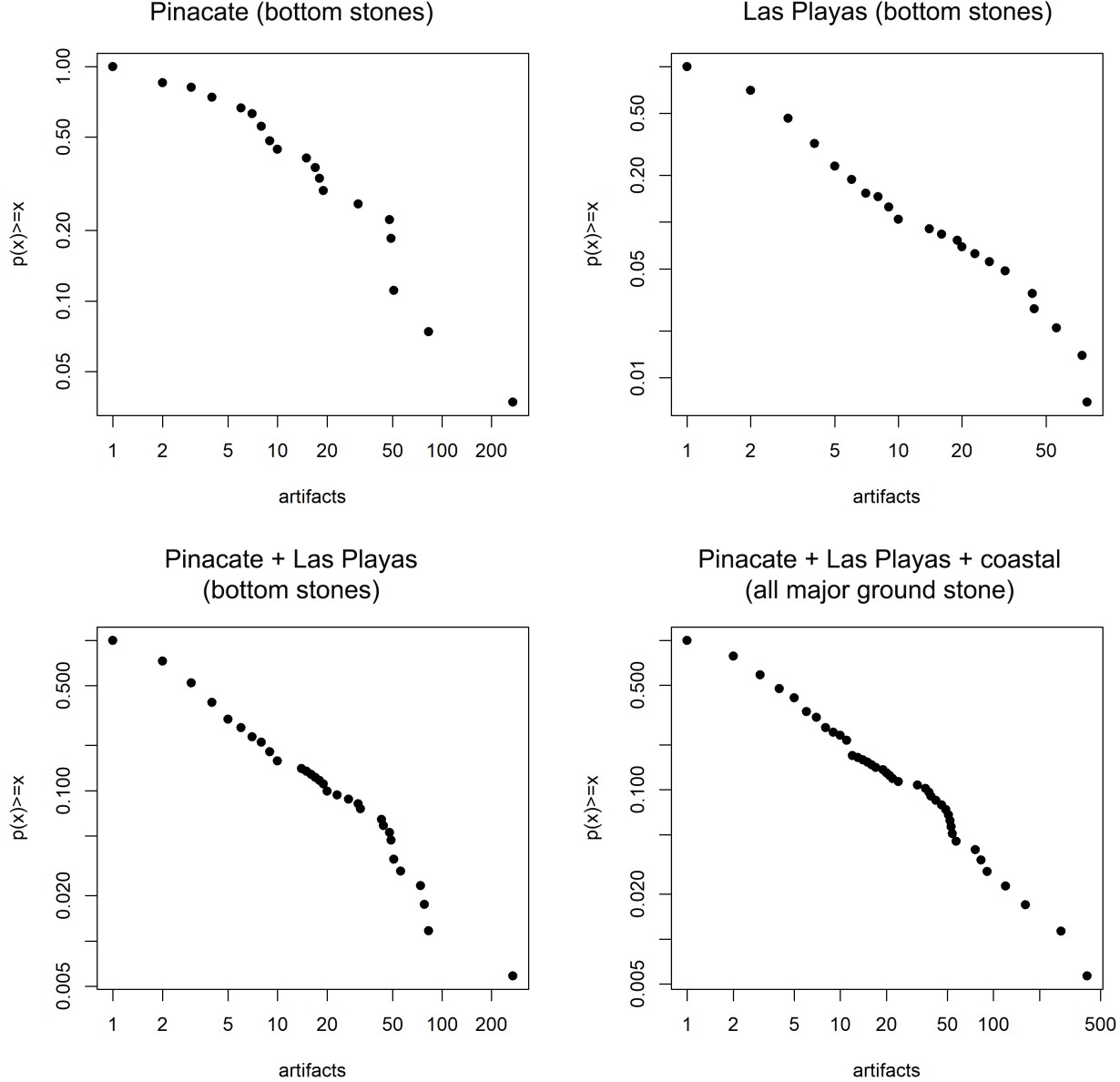

**Fig 5. Cumulative mass function plots.** Log-linear structure commensurate with power-law distributions is evident. Note linearity generally improves with inclusivity of sample. See the S1 Supporting Information in S1 File for caveats pertaining to the Coastal sample inclusive plot.

values is drawn from the MLE statistical model to produce a synthetic data set that is iterated 2500 times to measure the proportion of $D_s$ greater than $D_m$. As proposed by Clauset et al. [59] and implemented by Haas et al. [29], results with p > .1 are deemed plausible fits for a given distribution. Because real world phenomena cannot be scale free across an infinite range, we also explore the potential for power-law character only in the upper tail of distributions. This requires applying an iterative process to find the most-probable threshold value of $x_{min.}$ Once this parameter is defined, the above described MLE approach is applied to again test for plausibility. Due to zero values being problematic, we follow established protocols to add a value of 1 to all values in the dataset prior to analyses. This is justifiable based on the logic that no site would be recorded and thus in the dataset if it did not represent some interval of human time spent at the location.

Thus, zero values for ground stone are an issue of sampling through an imperfect proxy rather than representing true zeros which would be analogous to no time spent at the location. These explorations confirm power-law structures as a plausible fit among the tested distributions for both sub-sets of the sample and the overall sample of the combined Sierra Pinacate and Las Playas sites (Table 1), but notably this structure is only reliably perceptible in the upper tails of distributions. These results are very similar to the results obtained by Haas et al. [29] applied to projectile points.

## Relationship between occupation intensity proxies

We now turn to the primary application of exploring alternative proxies of occupation intensity to investigate variability among the sites of the corresponding settlement patterns. On first principles, we expect a strong positive correlation between site size and the number of ground stone or number of structures. Deviations of various magnitudes from this expected relationship can help us differentiate between the various components that make up occupation intensity, i.e., number of occupants, average duration of use, and site reuse frequency. Both ground stone and structure counts show only a weak relationship with site area (Table 2). This is true regardless of whether the data are employed in their raw format or are log-transformed. This corresponds closely to the insights of Haas et al. [29] but contrasts with applications that argue area is a reliable indicator of maximum population as indicated by close correlation with habitation features [19,24–26]. We find no flaws with these prior applications and stress the departure is almost certainly due to the frequent reuse of sites in the Pinacate region that confounds the relationship between site area and an inclusive definition of occupation intensity. However, we stress this is the far more common character of settlement pattern data pertaining to mobile groups.

Ground stone and structure counts are strongly correlated with each other in non-transformed data. The relationship is less strong in log-transformed data, but still statistically significant. This strong correlation between habitation features and artifact counts is a significant departure from prior investigations of mobile groups [24] and suggest in the Pinacate case both reflect accumulations over time rather than the momentary maximum of occupation as is sometimes inferred for feature data. Ground stone implements should have much longer use lives than wind break structures on the order of years or decades compared to days to seasons. The fact that a linear relationship holds between structures and ground stone indicates the population of wind breaks was not wholly recycled/reused with every site visit, allowing the population to grow over time. The faster growth in ground stone relative to structures is in keeping with our basic understanding of

**Table 1. Statistical results of the distribution fitting process.**

| Vector description | Model | Alpha | n | Prob | D | lambda | Xmin | p | log likelihood |
|---|---|---|---|---|---|---|---|---|---|
| Pinacate Las Playas combined | Power-law | 1.59 | 171 | | 0.16 | | | 0 | −474.16 |
| Pinacate | Power-law | 1.36 | 27 | | 0.23 | | | 0.01 | −115.05 |
| Las Playas | Power-law | 1.67 | 144 | | 0.17 | | | 0 | −354.45 |
| Pinacate Las Playas combined | Power-law tail | 1.88 | 89 | | 0.05 | | 3 | 0.48 | −284.14 |
| Pinacate | Power-law tail | 1.75 | 18 | | 0.13 | | 6 | 0.3 | −77.95 |
| Las Playas | Power-law tail | 1.99 | 101 | | 0.05 | | 2 | 0.46 | −250.90 |
| Pinacate Las Playas combined | Poisson | | 171 | | 0.65 | 9.16 | | 0 | −2126.92 |
| Pinacate | Poisson | | 27 | | 0.67 | 26.81 | | 0 | −750.21 |
| Las Playas | Poisson | | 144 | | 0.52 | 5.85 | | 0 | −976.66 |
| Pinacate Las Playas combined | Geometric | | 171 | 0.10 | 0.30 | | | 0 | −558.71 |
| Pinacate | Geometric | | 27 | 0.04 | 0.26 | | | 0.004 | −116.30 |
| Las Playas | Geometric | | 144 | 0.15 | 0.27 | | | 0 | −409.97 |

[a]Value of 1 added to all assemblage totals, to remove 0 values.

**Table 2. Correlation coefficients for loci proxies of occupation intensity in the pinacate sample.**

| | | Area (m$^2$) | Total structures |
|---|---|---|---|
| **Loci r$^2$ raw data** | | | |
| | Area (m$^2$) | | |
| | Total structures | 0.0225 | |
| | All bottom stones | 0.0081 | 0.7744 |
| **Loci r$^2$ log transformed** | | | |
| | | ln(area m$^2$) | Total structures |
| | ln(area (m$^2$)) | | |
| | ln(total structures) | 0.0337 | |
| | ln(all bottom stones) | 0.1198 | 0.205 |

The top portion of the table presents the correlation coefficients for the untransformed variables and the bottom portion presents the correlation for the log transformed variables (value of 1 added to count data prior to log-transform).

how these material proxies operated in culture systems. Previously used wind breaks would be periodically retired and removed from the record as their foundation rocks were repurposed or naturally/anthropogenically dispersed. Ground stone does not "decay" in this manner and actually "grows" as discarded specimens become fragmented. The relatively consistent relationship between ground stone and structures is the basis of several important insights: 1) the proxy data is internally consistent at the landscape scale; 2) larger settlements must have been subject to physical drift through time facilitated by frequent reuse to allow structures to accumulate, and 3) sites that deviate strongly from the relationship between ground stone and structures served special or more limited functions.

### Interpreting disparities in occupation intensity proxies

At this point in the analysis, we have demonstrated that ground stone scales as predicted by theory and prior empirical observation of hunter-gatherer settlement systems with a plausibly power-law distribution and that ground stone and structure count proxies have potential to capture complementary aspects of occupation intensity. We can now further explore these proxies' relationship to site size and each other to facilitate a finer scale interpretation of the relevant hunter gatherer settlement system. Exploring first the relationship between proxies and site size, it is apparent that the poorest correlations are achieved with the Pinacate sample and generally improved with the addition of the Las Playas and Coastal site data (S2 Table in S1 File) (Figs 5, 6). This observation captures the fact that most of the Las Playas sites have relatively simple occupation histories as either short term base camps in a residential mobility mode or logistical camps in a logistical mode. Small sites certainly show evidence of reuse, but it was much less frequent and apparently retained a consistent relationship between ground stone deposition and site size. This could be achieved by either adding new ground stone implements to newly occupied adjacent space or occupying the same space and reusing the same ground stone through time – a feasible strategy for relatively short stays. The Coastal data is more limited and we are reluctant to provide sweeping interpretations, but it appears many of these sites grew through accretion around expansive resource extraction areas—they added space and materials proportionately as they were not tied to spatially constrained resources. More interesting to our analysis is the observation that more logistically organized areas, such as the ethnographically described Pinacate tinaja adjacent camps, sharply depart from a reliable relationship between area and material proxy. As noted above, ethnographic studies that forward settlement area as a reliable proxy of occupation intensity largely focus on samples or sub-samples of hunter-gatherer sites corresponding to high residential mobility. This selectivity captures a less dynamic set of factors for site use than what is present in the totality of most hunter-gatherer systems [28], particularly those on the logistical end of the spectrum or those that seasonally alter their position on this continuum. The Pinacate results clearly support the observations by Haas et al. [29] that site area is a poor proxy for occupation

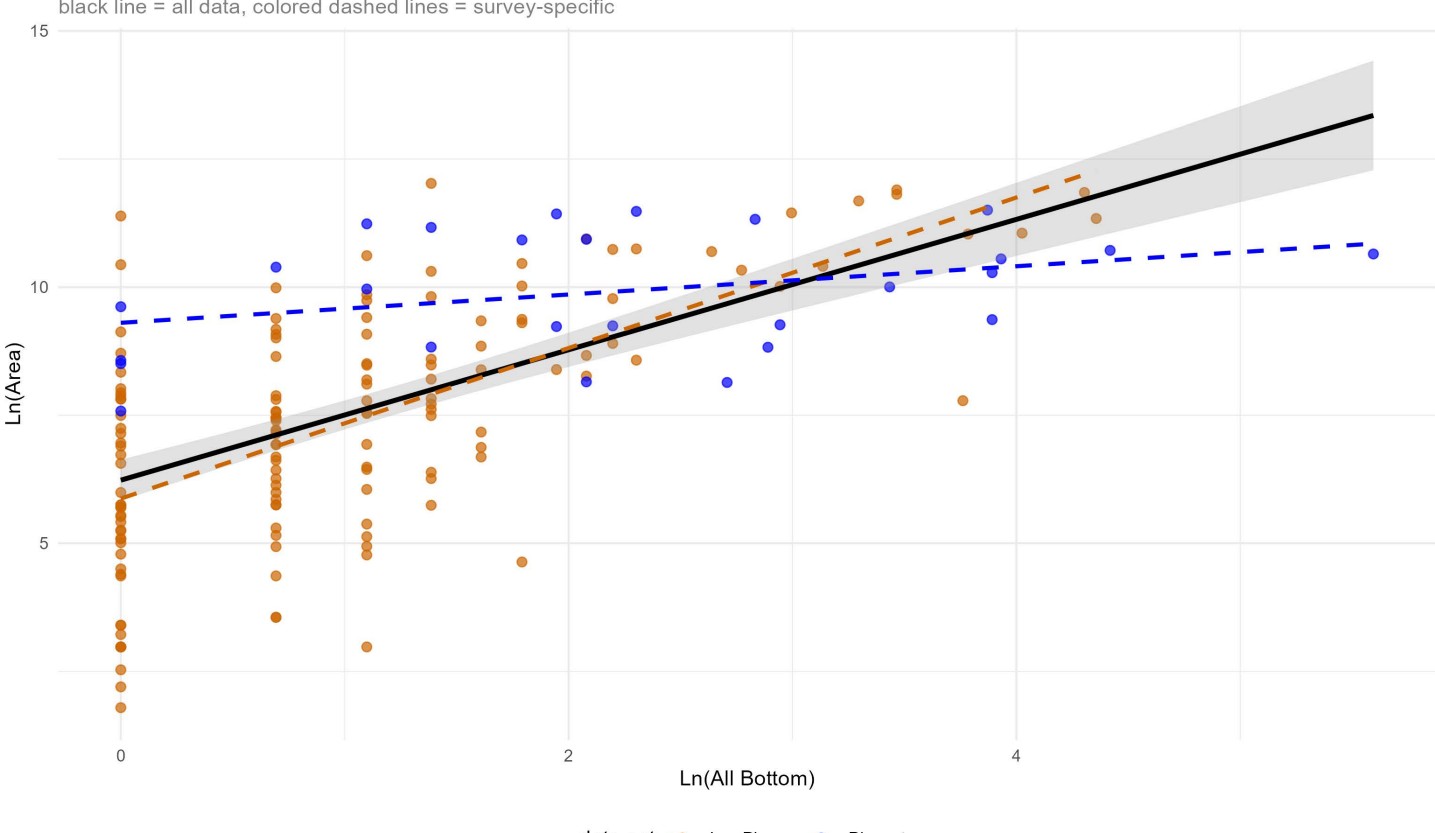

**Fig 6. Regression analysis of survey data.** The plot visually demonstrates the much improved fit between area and count of bottom stones of the Las Playas data set compared to the Pinacate data set. This data is log-transformed.

intensity but adds the qualification that this is predominantly driven by the presence of frequently reused logistical base camps that complicate the relationship between site area and material proxy.

These observations can also be evaluated through the empirically generated baselines of Settlement Scaling analysis. Briefly, substantial theoretical and empirical evidence provides benchmark estimates for the scaling factor expected between habitation sizes and social rates. As it relates to demographics and their proxies, there is a tendency for densification as settlements grow with the exponent in the formula $Y = aX^{\beta}$ falling in the range from 2/3–5/6, where Y is settlement size and X is population proxy. When data are log-transformed, this formula can be rendered as $Y = \beta \log X + \log a$ in which β is derivable from the slope coefficient from a regression model. Importantly, as mentioned above, hunter-gatherers typically violate the assumptions of Settlement Scaling Theory in that they require more space per person as populations grow [26], producing exponents >1. The critical point for our analysis is that substantial prior investigations provide a baseline of expectation for hunter-gather settlements in which the settlement size is observed to correspond to the momentary maximum population based on the actual physical space occupied (not the extent of space occupied cumulatively between uses). Substantial deviations with exponents (slopes) <.66 thus reflect site histories in which the material correlates of occupation intensity accumulated across many visits as opposed to singular visits. For our Pinacate sample, with exponents (slopes) that are well below.66 (S2 Table in S1 File) we thus have confirmation that reuse is a principal structuring component of occupation intensity. Commensurate with our above observations, the larger sample

that include the Las Playas or Las Playas and Coastal regions produce exponents much closer to those expected from ethnographic estimations of hunter-gatherer contexts [19,24,25]. This again reflects the overall residential mobility character of these areas and highlights the notable departure of the Pinacate portion of the settlement system. Tentatively, we suggest these already widely utilized approaches have unrecognized potential for identifying the relative contributions of site reuse in total occupation intensity.

Our Pinacate sample is not composed entirely of long term logistical base camps and this is readily apparent in the groupings of sites on the scatter plot diagrams and in the above distributional results. The sites clustered in the left corner of Fig 7 and Fig 8 are almost all short term camps as evidenced by small site sizes with few features or artifacts. To the right, sites with low ground stone/feature counts form a loose cluster spanning a significant size range that reflect locations of less persistent water, but nonetheless periodically predictable tinajas. Both the ecological parameters and archaeological data suggest these locations were probably visited frequently but only for short periods by small groups. These were reoccupations of the same general area, producing very diffuse artifact and feature scatters. In these cases, we infer that occupants planned for only short term stays, leading to a minimal structuring of space maintained between visits, as the degree of site organization is often linked to the planned (rather than actual) duration of occupation [2,61]. In several cases, ground stone was clearly reused frequently between visits, such as the very deep mortars at Galletal further

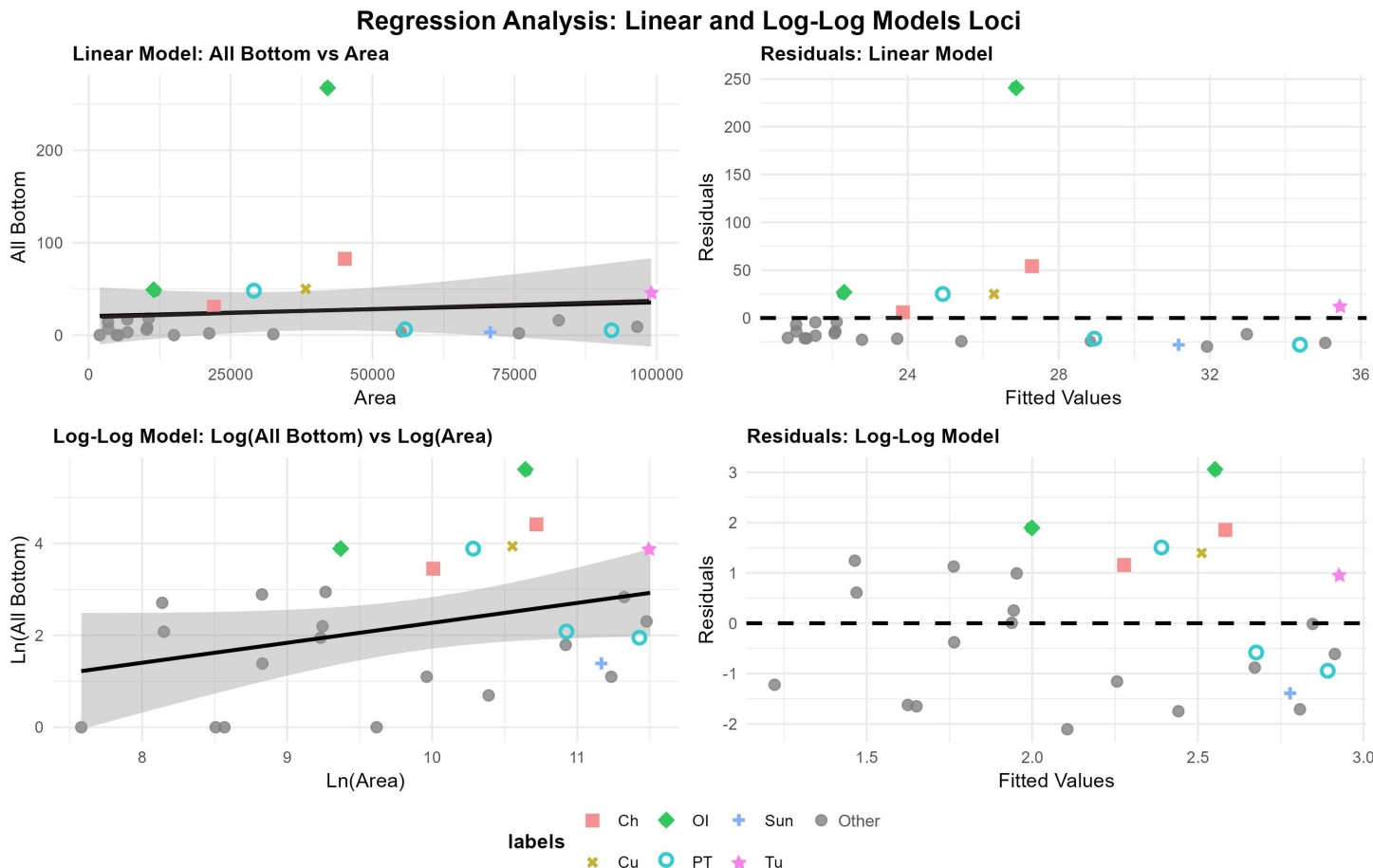

**Fig 7. Linear and log-log regressions with residuals vs. fitted values.** This plot explores the relationship between site area and count of bottom stones. Abbreviations: Ch-Chivos, Cu-Cuervos, OI-Ojo de Iitoi, Papago Tanks-PT, Sun-Sunset, NA-all other sites.

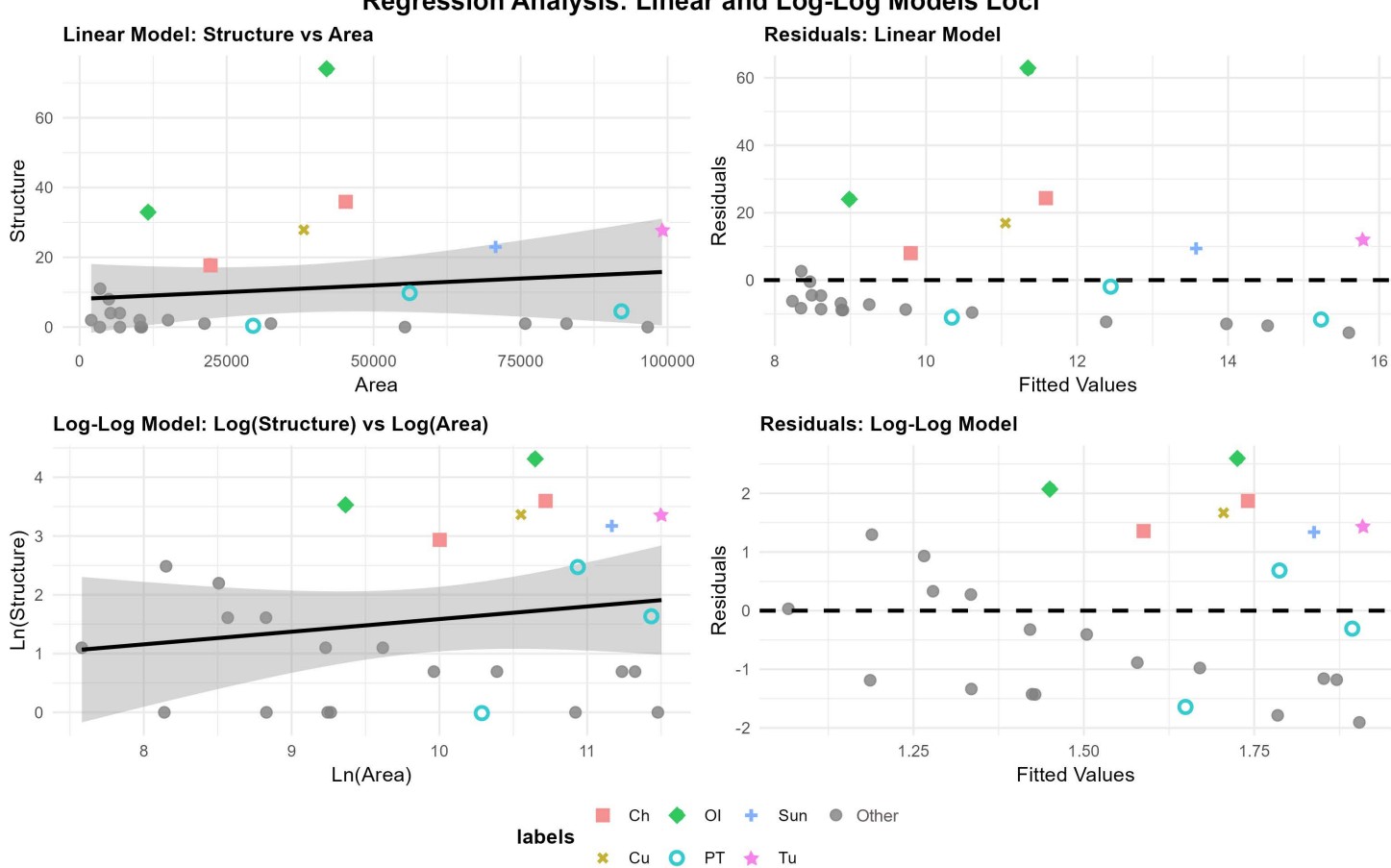

**Fig 8. Linear and log-log regressions with residuals vs. fitted values.** This plot explores the relationship between site area and count of structures. Abbreviations: Ch-Chivos, Cu-Cuervos, OI-Ojo de Iitoi, Papago Tanks-PT, Sun-Sunset, NA-all other sites.

separating the sites from the regression line—size grew while artifact inventory did not. The sites above the regression line (zero line on residual plots) correspond almost exactly with those identified as long-term logistical base camps tied to the key resource of reliable tinajas. We have ethnohistoric confirmation of this character for several examples and for all there is obvious archaeological evidence especially in the case of the most extreme outlier of Ojo de Iitoi (loci 3). Among these logistical base camps, the area data are decidedly noisy. We infer the magnitude of of the positive residual to reflect the relative contribution of site reuse and possibly also average duration of stay. In other words, the longer the average duration of individual occupations and greater the frequency of substantial site reuse (multi-annual in some cases) the greater the disproportionality in artifact and feature accumulation relative to area. Ojo de Iitoi, and specifically loci 3, presented artifacts from the Early Archaic to the Historic era indicating the overall period of use (number of reuse events) is a strong driver of the degree of departure from the "expected" site area.

Finally, we turn our attention to the notable departures between the overall reliable relationship between the material proxies of structures and ground stone. Again we present the data in both raw and log-transformed formats and with plots of residuals (Fig 9). The site of Sunset occupies a notably positive residual position in this relationship owing to its relative lack of ground stone. This was the only substantial site recorded that was not located adjacent to a tinaja. This site is described as the "capital" of the Hia-Ced for its role in ceremonial practices [62:162]. Occupation

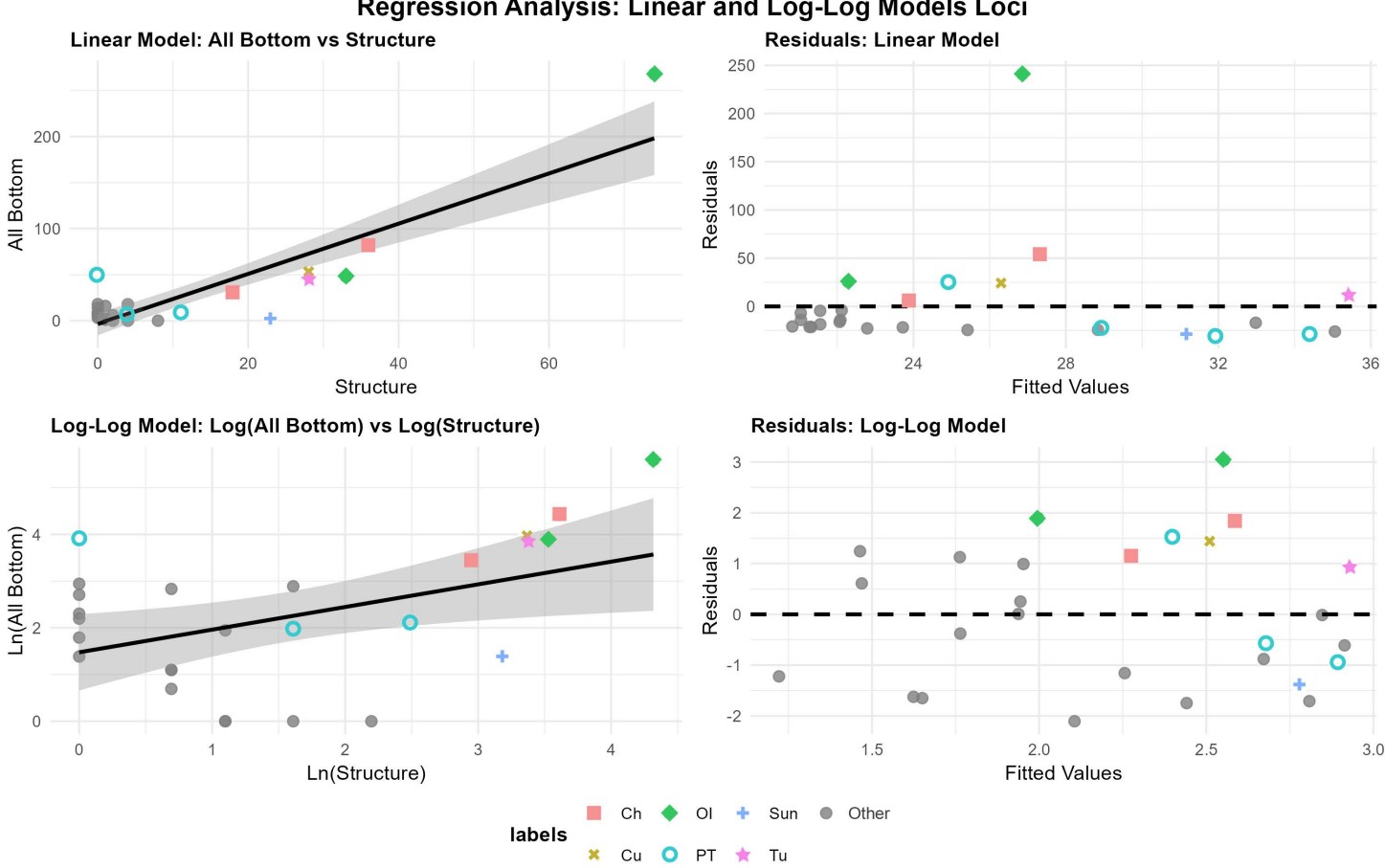

**Fig 9. Linear and log-log regressions with residuals vs. fitted values.** This plot explores the relationship between count of structures and count of bottom stones. Abbreviations: Ch-Chivos, Cu-Cuervos, OI-Ojo de Iitoi, Papago Tanks-PT, Sun-Sunset, NA-all other sites.

at this site, though incorporating domestic activities, clearly was undertaken with an expectation of impending future mobility that discouraged substantial investment in site furniture (ground stone). Unlike the other short term use sites, the social goals underlying occupation required a much more moderated use of space, resulting in a far less diffuse dispersion of artifacts. The notable ground stone to structure ratio thus captures the ethnographically confirmed special ritual function of this site.

In contrast, the negative residual position of Papago Tanks stands out for its lack of structures. The spatial partitioning of features pertaining to different domains of activity suggest this location is best considered as an integrated site, but we maintain our loci level analysis for consistency. Loci 1, dominated by numerous grinding slicks and bedrock mortars adjacent to the tinaja is completely lacking in structures. The few structures present at Papago Tanks are predominantly located along an intersecting arroyo, loci 3. One of the largest features discovered on the project, a large rock oval with a cleared desert pavement center is located in loci 2, which otherwise includes only very dispersed artifacts and features. Loci 1 and 2, and the whole site when considered at this scale (S1 Supporting Information, S4 Fig in S1 File) occupy a notably negative residual position. Ethnohistoric texts identify Papago Tanks as the most reliable tinaja in the region. Located on the northern margin of the Sierra Pinacate, it was accessible by groups traveling between different ecological zones. The location appears to have served as a common pool resource shared

by multiple groups and was so highly valued for its reliable water that groups would likely engage in considerable violence to maintain access. To the south in the equally arid Comca'ac region ethnographic accounts note groups did not establish camp locations in the immediate surroundings of water holes to avoid laying claim and thus inciting intra-ethnic conflict [47]. We suspect Papago Tanks, like several other locations in the Sonoran Desert [48], also served diverse inter- and intra-ethnic groups through similar social mechanisms applied to common pool resources. The lack of structures except in the more distant loci 3 is readily interpretable in this framework. The fact that the most prevalent ground stone are bedrock slicks and mortars, which cannot be carried away by hostile groups is also suggestive. Finally, the prevalence of ground figures including the large oval feature is suggestive of efforts at group integration. Where better to hold such events than at a place where it is most necessary to demarcate group boundaries through ritual as a means of resource access? This interpretation accords well with the ethnographic record, but importantly would be inferable from purely archaeological and environmental evidence.

## Conclusion

This investigation applied broad theoretical insights to an ethnohistorically informed case study. The effort offers confirmatory evidence of generalizable precepts and further advances investigations of disparities between site size and material proxies of occupation intensity. Ethnographic and archaeological data pertaining to hunters-gatherers have diverged on the utility of site size to serve as a proxy of occupation intensity. Our data suggests this is at least in part a reflection of the investigated data sets. Those corresponding to groups with high residential mobility with less tendency to reuse the same camps or that employ methods or proxies that only capture the momentary maximum population are far less susceptible to a critical dimension that structures the vast majority of archaeological site data. That power-law relationships are perceptible in both ethnographic patterns and archaeological patterns masks the fact that the underlying causality of the distributions is not invariant. Our data offer additional confirmation that power-law structure is also captured in the material remains of logistically organized systems. This is not surprising as examining only the forager component of a seasonal round would be expected to exclude the largest upper-tail sites that would result from longer term occupation of logistical base camps. Building on insights from Settlement Scaling Theory, our data suggest that the value of the slope of a log-transformed plot of material proxies against site size may provide an easy means of identifying settlement patterns significantly affected by site reuse.

Our most significant and novel contribution comes through exploring the relationship between multiple proxies of occupation intensity in relation to site area. The departures from this correlation provide a more refined view of how individual sites and groups of sites fit into larger patterns of landscape use. Though we marshal ethnohistorical reports of site use to verify relationships in these proxies, particularly in the upper tail of feature/artifact counts, our interpretations would have been inferable without this data. Results conform to predictable patterns of landscape use, ecological factors, and social systems of hunter-gatherers. Some of the most interesting departures concern behaviors meant to alleviate inter-group tensions or build intra-group social cohesion. Indeed, two of the largest departures invoke a specialized ritual nature of the site, as further verified by preserved features in one case, thus providing rare support for the importance of this often invoked but little demonstrated realm of behavior.

## Supporting information

**S1 File. Supporting information, including S1 Supporting Information, S1 Fig, S2 Fig, S3 Fig, S4 Fig, S1 Table, S2 Table, and S3 Table.**
(DOCX)

**S2 File. Inclusivity in global research questionnaire.**
(DOCX)

## Acknowledgments

We are indebted to community members Cristina Lizarraga and Israel Barba for their efforts in facilitating our project. Thank you to Ben Wilder and Jay Quade for their assistance in facilitating the Pinacate survey. Ben Wilder provided helpful comments on the text as did three anonymous reviewers. This effort is dedicated to our Project Director John Carpenter who passed away in 2024.

## Author contributions

**Conceptualization:** Amy E. Clark, Guadalupe Sánchez Miranda, Matthew Pailes.

**Data curation:** Neftalí López-Pérez.

**Formal analysis:** Guadalupe Sánchez Miranda, Neftalí López-Pérez, Antonio López-Rivera, Matthew Pailes.

**Funding acquisition:** Guadalupe Sánchez Miranda.

**Investigation:** Guadalupe Sánchez Miranda, Antonio López-Rivera, Tamara Luna, Astrid Avilés, Richard Martynec, Sandra Martynec, Natalia Martínez-Tagüeña, Matthew Pailes.

**Methodology:** Amy E. Clark, Neftalí López-Pérez, Antonio López-Rivera, Natalia Martínez-Tagüeña, Matthew Pailes.

**Project administration:** Guadalupe Sánchez Miranda, Antonio López-Rivera, Tamara Luna, Astrid Avilés, Richard Martynec, Sandra Martynec, Natalia Martínez-Tagüeña.

**Resources:** Neftalí López-Pérez, Richard Martynec.

**Supervision:** Amy E. Clark, Guadalupe Sánchez Miranda.

**Visualization:** Neftalí López-Pérez.

**Writing – original draft:** Amy E. Clark, Matthew Pailes.

**Writing – review & editing:** Amy E. Clark, Guadalupe Sánchez Miranda, Richard Martynec, Sandra Martynec, Matthew Pailes.

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
