## [Decision Letter · Decision Letter 0]

17 Apr 2025

Dear Dr. Pailes,

Thank you for submitting your manuscript to PLOS ONE. After careful consideration, we feel that it has merit but does not fully meet PLOS ONE’s publication criteria as it currently stands. Therefore, we invite you to submit a revised version of the manuscript that addresses the points raised during the review process.

We look forward to receiving your revised manuscript.

Kind regards,

Briggs Buchanan, Ph.D.

Academic Editor

PLOS ONE

Journal Requirements:

3. In your manuscript, please provide additional information regarding the specimens used in your study. Ensure that you have reported human remain specimen numbers and complete repository information, including museum name and geographic location.

For more information on PLOS ONE's requirements for paleontology and archeology research, see https://journals.plos.org/plosone/s/submission-guidelines#loc-paleontology-and-archaeology-research .

“GS; 2021-007; Agnese Nelms Haury Charitable Trust of the University of Arizona; https://www.haury.arizona.edu/

GS; NA; Centro-INAH-Sonora; NA”

“We are indebted to community members Cristina Lizarraga and Israel Barba for their efforts in facilitating our project. Ben Wilder and Jay Quade obtained the initial funding that supported this research. This effort is dedicated to our Project Director John Carpenter who passed away in 2024.”

“GS; 2021-007; Agnese Nelms Haury Charitable Trust of the University of Arizona; https://www.haury.arizona.edu/

GS; NA; Centro-INAH-Sonora; NA”

7. Please include a caption for figure 6 and 7.

8. Please include captions for your Supporting Information files at the end of your manuscript, and update any in-text citations to match accordingly. Please see our Supporting Information guidelines for more information: http://journals.plos.org/plosone/s/supporting-information .

Reviewers' comments:

Reviewer's Responses to Questions

**Comments to the Author**

1. Is the manuscript technically sound, and do the data support the conclusions?

Reviewer #1: Partly

Reviewer #2: Partly

2. Has the statistical analysis been performed appropriately and rigorously?

Reviewer #1: Yes

Reviewer #2: No

3. Have the authors made all data underlying the findings in their manuscript fully available?

Reviewer #1: Yes

Reviewer #2: Yes

4. Is the manuscript presented in an intelligible fashion and written in standard English?

Reviewer #1: Yes

Reviewer #2: Yes

Reviewer #1: The authors tackle an interesting issue in the analysis of hunter-gatherer settlement data. In many agrarian systems site area has a strong relationship with population proxies. But in H/G systems, site area is less well correlated with other measures of occupational intensity. The authors find the same here and discuss possible explanations for it.

There is a huge literature that examines the relationship between momentary population and area across permanent settlements, and the patterns are very clear there, as exemplified by Ortman et al. 2020. Discussing the regularity of these patterns would help to motivate the current work with hunter-gatherer data.

Regarding the distributional fits, I would think testing for a log-normal distribution would also be appropriate given that the upper tails of power and log-normal distributions are often very similar and difficult to distinguish from each other. I would add that to the list of MLE tests.

I was also surprised not to see comparisons between area and features/artifacts using scaling analysis, with linear fits to log transformed variables. This is the more standard approach in the recent literature, and it allows one to account for nonlinear correlations between variables.

I couldn’t tell from Table 1 and the associated text which of the datasets is being used there. I think it must be Pinacate because these are the only data in the appendix that have area, structures, and groundstone tabulated at both the site and locus level. This should be clarified.

The scatterplots are also pretty drafty looking and need to be improved for legibility.

Once I looked at the data in the associated zip file I understood the analyses much better, but I have to admit that in reading the paper I often had difficulty telling which data were being discussed in which case. In fact, it took me awhile to even realize that there were different subsets of data that were being analyzed in different cases. It would be good for the authors to work on clarifying the language and presentation to make that more clear.

The relationships between total structures, and total ground stone, vs. site area are really different depending on which dataset is used. For example, in their all regional dataset, the scaling relationship between total groundstone and total area (slope of the fit line on a log-log plot) has a slope of about 1; but in their Pinacate loci dataset, even when I add together all structures and all groundstone and compare that to site area, I get a slope of about .25 in the scaling relationship. So, in the regional data, site area increases roughly proportionately with the amount of material on the site, but in the Pinacate data the amount of material increases much faster than the area.

It would be worthwhile for the authors to put a bit more effort into comparing their results with other studies of hunter-gatherer settlement data. In addition to the study by Haas et al. that the authors focus on, Lobo et al. and Hamilton et al. look at the relationship between population and area across ethnographic camps, and Ortman et al. examine the relationship between stone circle count and area in Wyoming stone circle sites. There is variation in these relationships across cases, but the slope is never as low as .25.

One simple possible explanation is site boundary definitions, such that density thresholds that define site edges decrease as the site size increases. It would be good to rule this out somehow.

Alternatively, one might not expect total structures or artifacts to be proportional to the average momentary populations of sites, and the authors acknowledge this by structuring their discussion around occupational intensity instead of population per se. Something that occurs to me is that in this system more areally-extensive sites might be used by more people but much less frequently, during periodic aggregations only when conditions permit; whereas smaller sites might be used much more frequently or for longer durations by smaller groups. The density of ground stone definitely decreases with site area in the all regional data, so overall larger sites are used less intensively. This would seem to imply that smaller sites were used MUCH more often than larger sites. I suppose it is also possible that the accumulation rate of features and artifacts per person day of use decreases as the group size increases, as might occur if large groups gathered only infrequently and for short times relative to smaller groups. Lobo et al. discuss a model along these lines that might well fit this resource limited environment, where there is an intrinsic energy deficit involved in camping together in large groups, but people still want to gather when they can for social production, courtship, trade, information sharing, etc. Features and groundstone might then accumulate at slower rates per person day in larger sites because people are living with an energy deficit when camping together in large groups. These are just ideas for the authors to consider.

Regarding the distributional fits, I would think testing for a log-normal distribution would also be appropriate given that the upper tails of power and log-normal distributions are often very similar and difficult to distinguish from each other. I would add that to the list of MLE tests.

Overall, I think this is an interesting dataset and I appreciate the authors’ careful conceptualization of the factors involved in the variation observed. In addition to considering the reactions above, I think the main thing the paper could use is a bit more organizational work on the discussion of the actual analyses. The background material is interesting, but the analysis write up is harder to follow. I could eventually figure it out, but I am putting more work into the paper as a reviewer than the average reader will.

Citations

Ortman, Scott G., José Lobo and Michael E. Smith 2020 Cities: Complexity, theory and history. PLOS ONE 15(12):e0243621.

Lobo, José, Todd Whitelaw, Luís M. A. Bettencourt, Polly Wiessner, Michael E. Smith and Scott Ortman 2022 Scaling of Hunter-Gatherer Camp Size and Human Sociality. Current Anthropology 63(1):68-94.

Ortman, Scott G., Laura L. Scheiber and Zachary Cooper 2022 Scaling analysis of prehistoric Wyoming camp sites—implications for hunter-gatherer social dynamics. In Intra-Site Spatial Analysis of Mobile and Semisedentary Peoples: Analytical approaches to reconstructing occupation history, edited by A. E. Clark and J. A. M. Gingerich. University of Utah Press, Salt Lake City.

Hamilton, Marcus J., Briggs Buchanan and Robert S. Walker 2018 Scaling the Size, Structure, and Dynamics of Residentially Mobile Hunter-Gatherer Camps. American Antiquity 83(4):701-720.

Reviewer #2: In this study, Clark et al investigate how well different archaeological proxies reflect occupation intensity in hunter-gatherer settlement systems, using data from the traditional homeland of the Hia-Ced O’odham in the Sierra Pinacate and adjacent regions of Sonora, Mexico. Drawing on both archaeological survey data and ethnohistoric records, the authors assess how features such as site size, ground stone artifact counts, and windbreak structures relate to occupation intensity, conceptualized as a combination of group size, duration of stay, and frequency of reuse.

They find:

1) Ground stone counts and structure counts are robust proxies for occupation intensity, whereas site area shows only a weak and inconsistent relationship.

2) Occupation intensity across the landscape follows a power-law distribution, consistent with theories of preferential site reuse.

3) Divergences between site size and material proxies are not seen as confounding, but rather as analytical opportunities to infer site-specific roles in broader settlement systems.

4) Ethnohistoric insights help interpret these divergences, revealing factors like ritual use, common-pool resource dynamics, and inter-group cooperation/conflict that influenced site use.

By combining detailed regional archaeology with theoretical modeling and ethnohistoric context, the authors suggest that deviations in proxy relationships can yield nuanced reconstructions of landscape use, especially in systems featuring mixed forager-collector strategies.

The data are interesting and some of the results are quite suggestive (see Figure 5, for example). Others are less compelling. The correlations in Figure 7 are hard to interpret as the majority of the data is located to the lower left quadrant, and most clustered near zero. This both anchors the data artificially and increases undue leverage as we move up the x and y axes. The data point in the upper right (Od13) clearly has undue leverage here. This means the data need to be logged (and then you will find similarly strong, but more robust results).

There are also minor issues with terminology: for example, in line 547 the authors state “The site of Sunset 547 occupies a notably supra-linear position in this relationship” and in line 563 “Papago Tanks as individual loci or a unified site, it retains a 564 notable sub-linear status”. The authors here are referring to positive or negative residuals. The terms supra- or sub-linear refer to linearity of the fits (i.e., the slopes of regressions), not to the deviation of individual data points from their expected values.

My more major concern with the manuscript in this draft is the clarity of the writing. The paper starts quite strongly with a solid introduction to the issue of hunter-gatherer mobility (but there are more papers to cite here) and highlights the lack of direct archaeological proxies. This is great and unquestionably true.

However, once the authors get into the Data Collection Methods and Goals and beyond, the paper becomes increasingly hard to follow. Different kinds of data were collected from different sites (this seems justifiable) but it is difficult for the reader to follow this, and the variables are given entirely non-intuitive names “all bottoms”, “sum bottoms”, “all ground stone”. As such, it is very hard to keep these straight. And this goes further into the Proxy Quantification section. By the time we reach Survey Results, I am already confused.

It is also unclear why the Analysis and Discussion sections are lumped together. The authors start with a discussion of the workflow and present results, which seem to be interspersed with interpretations, caveats, and qualifiers. Again, this makes it hard to follow.

The section Assessing the representativeness of the settlement pattern I found particularly hard to follow. As someone familiar with these types of analysis, I found the presentation a little convoluted and confusing. In particular, I did not follow lines 474 to 485. There are various statistical methods of dealing with zero-inflated data, but, although widely practiced, adding “1” to all values when zeros are real data has no statistical meaning, especially when dealing with data on these logarithmic scales. By the time we reach the Conclusions, I was not overwhelmed by their support from the data, or exactly what I should have learned.

In sum, many of the individual sections need serious editing and trimming as they are not clear. Mu suggestion would be to simply focus of the strongest results in Figure 5 and tell us about their importance. Remove much of the other analysis and description to Supplementary Materials. At the moment, there seems to be a very simple and perhaps compelling story to tell here from this case study, but currently that message is buried under quite an opaque text.

**Do you want your identity to be public for this peer review?** For information about this choice, including consent withdrawal, please see our Privacy Policy

Reviewer #1: No

Reviewer #2: No

---

## [Author Response · Author response to Decision Letter 1]

2 Sep 2025

This information was uploaded as a separate document.

---

## [Editor Report · Decision Letter 1]

22 Sep 2025

Exploring proxies for occupation intensity in hunter-gatherer settlement systems: A combination of ethnohistoric and archaeological data

PONE-D-25-11723R1

Dear Dr. Pailes,

We’re pleased to inform you that your manuscript has been judged scientifically suitable for publication and will be formally accepted for publication once it meets all outstanding technical requirements.

Kind regards,

Briggs Buchanan, Ph.D.

Academic Editor

PLOS ONE

Additional Editor Comments (optional):

The revised manuscript is much improved and ready for publication. I would urge the authors to make a few minor changes suggested by reviewer #2 regarding the use of the terms sub- and supra-linearity, these terms denote properties of slopes not individual data points in this case specific sites, for examples on pages 30 and 31, using positive or negative residual or above or below the slope line is more appropriate. Congratulations on a terrific study.
---

## [Editor Report · Acceptance letter]

PONE-D-25-11723R1

PLOS ONE

Dear Dr. Pailes,

I'm pleased to inform you that your manuscript has been deemed suitable for publication in PLOS ONE. Congratulations! Your manuscript is now being handed over to our production team.

Kind regards,

on behalf of

Dr. Briggs Buchanan

Academic Editor

PLOS ONE